# Effects of Mars Global Simulant (MGS-1) on Growth and Physiology of Sweet Potato: A Space Model Plant

**DOI:** 10.3390/plants13010055

**Published:** 2023-12-23

**Authors:** Karthik Chinnannan, Prapooja Somagattu, Hyndavi Yammanuru, Padma Nimmakayala, Manohar Chakrabarti, Umesh K. Reddy

**Affiliations:** 1Department of Biology, Gus R. Douglass Institute, West Virginia State University, Institute, WV 25112, USA; karthik.chinnannan@wvstateu.edu (K.C.); psomagattu@wvstateu.edu (P.S.); hyammanuru@wvstateu.edu (H.Y.); padma@wvstateu.edu (P.N.); 2School of Integrative Biological and Chemical Sciences, University of Texas Rio Grande Valley, Edinburg, TX 78539, USA; manohar.chakrabarti@utrgv.edu

**Keywords:** Mars Global Simulant, stress tolerance, antioxidant, amino acids, proline, sugar classes

## Abstract

Growing food autonomously on Mars is challenging due to the Martian soil’s low nutrient content and high salinity. Understanding how plants adapt and evaluating their nutritional attributes are pivotal for sustained Mars missions. This research delves into the regeneration, stress tolerance, and dietary metrics of sweet potato (*Ipomoea batatas*) across different Mars Global Simulant (MGS-1) concentrations (0, 25, 50, and 75%). In our greenhouse experiment, 75% MGS-1 concentration significantly inhibited sweet potato growth, storage root biomass, and chlorophyll content. This concentration also elevated the plant tissues’ H_2_O_2_, proline, and ascorbic acid levels. Higher MGS-1 exposures (50 and 75%) notably boosted the vital amino acids and sugar groups in the plant’s storage roots. However, increased MGS-1 concentrations notably diminished the total C:N ratio and elemental composition in both the vines and storage roots. In summary, sweet potato exhibited optimal growth, antioxidant properties, yield, and nutrient profiles at 25% MGS-1 exposure as compared to higher concentrations. This study underscores the need for future interventions, like nutrient enhancements and controlled metal accessibility, to render sweet potato a suitable plant for space-based studies.

## 1. Introduction

Exploring life beyond our planet requires understanding the limits of life’s adaptability. The recent advances in space research have opened a new chapter in human exploration of Mars, as fueled by government and private space initiatives [1,2]. As the ambition to establish a human presence on the Red Planet intensifies, supplying essentials to astronauts will become a daunting task. The logistical hurdles, enormous costs, and inherent risks associated with continual Earth-to-Mars supply make it imperative to consider producing food directly on Martian soil. The soil of Mars, known as regolith, is a unique blend, abundant in both micro- (Cr, Cu, Fe, Mn, Mo, Ni, and Zn) and macro-nutrients (C, Ca, H, K, N, Na, Mg, O, P, and S) that are vital for plant growth [3]. However, its high concentrations of salts, specifically sulfates and perchlorates, coupled with heavy metals, render the soil inhospitable for vegetation. Moreover, the natural properties of Martian soil—scarce nutrient content, the presence of phytotoxic elements, excessive salinity, and inadequate water retention capacity due to the lack of organic matter—indicate that raw Martian regolith is not inherently suitable for farming [3,4,5]. Thus, identifying the optimal soil composition and understanding plant adaptability to Martian conditions is crucial for possible colonization.

A prior study compared the growth of wild plants, crop plants, and legume plants on lunar and Martian regolith simulants [6]. However, whether the simulants used in the experiment truly represent actual regolith remains questionable. Several other researchers have also ventured into studying the impact of Martian regolith on plant development, focusing on enhancing its nutrient content and experimenting with various soil combinations. These studies encompass aspects such as seed germination, plant morphology, and biochemistry across diverse plant species [2,3,7,8,9,10,11]. A standout study by Roberto et al. [12] identified specific genes and proteins in Italian ryegrass that respond to stress in Martian soil simulants. However, a comprehensive understanding of the nutritional characteristics under Martian conditions remains unachieved, highlighting the need to explore the dietary criteria for space sustenance.

Sweet potato (*Ipomoea batatas* (L.) Lam.), recognized as a dietary mainstay, ranks seventh among the worldwide food crops [13]. Its potential as a “space crop” is derived from its impressive nutritional content, efficient water utilization, rapid growth cycle, ease of processing, abundant yields, and extended storage life [14,15,16]. The adaptability of sweet potato is remarkable, as its vines are capable of regenerating new plants. Both its roots and tender leaves provide nutritious food sources [17,18]. A prior study demonstrated that, if supplemented with nutrient solutions, lunar and Martian regolith simulants can support the growth of sweet potato [11].

The majority of the Martian regolith simulants were developed around one or two decades ago. Our understanding of the physical and chemical properties of Martian regolith has increased significantly in recent years. In our experiment, we used MGS-1 (Mars Global Simulant-1), a newly developed, high-fidelity mineralogical analog of Martian regolith, that exhibits a remarkable advancement over the previously developed Martian regolith simulants [19]. Against this backdrop, our study was structured to (i) evaluate the adaptability and regenerative capabilities of sweet potato under various MGS-1 conditions, (ii) understand its stress resilience through biochemical and amino acid analyses, and (iii) determine the nutritional value of its storage roots when grown in MGS-1 environments.

## 2. Results

### 2.1. Effect of MGS-1 Exposure on Growth and Photosynthetic Pigments

Producing essential consumables using Martian soil presents a significant challenge in the colonization of Mars. As such, pinpointing a viable method for producing these consumables is paramount. This study evaluated the growth response of sweet potato to various MGS-1 exposure levels in order to understand its growth, stress adaptation, and nutritional attributes. Compared to the control group, different MGS-1 concentrations significantly hindered sweet potato growth and storage root formation. The most pronounced phytotoxic effects, including stunted growth, diminished leaf size, and leaf margin necrosis, were evident at 75% MGS-1 exposure. Notably, at 100% exposure, sweet potato slips failed to grow and perished within 3 to 5 days post-transplantation. Alongside this inhibited growth, there was a substantial reduction in the total storage root biomass by 40%, 66.53%, and 93.27% at 25%, 50%, and 75% MGS-1 exposures, respectively, when compared to the control group (Table 1 and Appendix A). Indicators like plant height and biomass serve as primary markers of phytotoxicity under abiotic stress conditions. These findings suggest that MGS-1’s phytotoxicity might stem from its sterility and its lack of biotic activity and essential nutrients. Simulated soils from extraterrestrial environments tend to lack vital nutrients for plant growth and development.

The total chlorophyll content in leaf tissue was analyzed to evaluate the physiological response of sweet potato to varying MGS-1 exposure levels. Spectrophotometric results mirrored the growth findings and showed a significant decline in the chlorophyll content of sweet potato leaves due to MGS-1 exposure. After 120 days of treatment, a peak chlorophyll content of 41.89 µg/mg FW was observed in the control leaves. This content decreased by 9.43%, 16.41%, and 5.37% at 25% exposure; 20.22%, 38.70%, and 15.61% at 50% exposure; and 46.43%, 64.4%, and 36.19% at 75% exposure for plants aged 40, 80, and 120 days, respectively (Figure 1A). Thus, the chlorophyll content was not affected much when treated with 25% MGS-1; however, the chlorophyll content was reduced significantly upon treatment with 50% or 75% MGS-1. This decline in chlorophyll content among the plants exposed to Martian regolith could be attributed to MGS-1 exposure inhibiting the enzymes vital for chlorophyll biosynthesis. Additionally, MGS-1 exposure reduced the total leaf area, possibly contributing to a decreased light harvesting rate and the subsequent impairment of photosynthesis.

### 2.2. Effect of MGS-1 Exposure on Antioxidant System of Sweet Potato

Proline serves as a recognized physiological marker for environmental stress, reflecting the plant’s level of adaptation to such stressors. In this study, exposure to MGS-1 was found to significantly influence proline accumulation, particularly by increasing the proline content in the leaves of sweet potato treated with MGS-1. When assessing the effects of various MGS-1 concentrations on proline accumulation, the most pronounced increases—33.56%, 60.83% and 100%–were observed at 50% exposure. This was followed by increases of 23.04%, 55.31%, and 86.17% at 75% exposure and 16.96%, 25.89%, and 59.82% at 25% exposure in plants aged 40, 80, and 120 days, respectively (Figure 1B). Interestingly, the proline accumulation trend in sweet potato was directly proportional to both MGS-1 concentrations and plant age up to 50% exposure and 80 days of treatment. After these points, the proline content started to decrease.

Environmental stress can profoundly disrupt the cellular redox balance, leading to the generation of reactive oxygen species (ROS) and subsequent oxidative damage in plants. In this context, this study examined the oxidative harm caused by MGS-1 by gauging the H_2_O_2_ accumulation. The prominence of the dark brown hues in leaf tissue served as a direct measure of ROS levels. The control leaf samples exhibited fewer dark brown spots across all three evaluated stages. Conversely, with increasing MGS-1 exposure, there was a progressive increase in H_2_O_2_ accumulation in the leaves, correlating with both the plant age and the levels of MGS-1. Notably, 120-day-old plants exposed to 75% MGS-1 demonstrated the highest accumulation of H_2_O_2_ as compared to other treatments (Figure 2A,B).

This study also revealed a concentration-dependent rise in the total ascorbic acid content in harvested sweet potato storage roots. The most substantial increase, 40.65%, was observed at 75% exposure, trailed by increments of 36.26% and 15.28% at 50% and 25% MGS-1 exposures, respectively (Table 1 and Appendix A). This surge in ascorbic acid might be attributed to sweet potato’s defensive mechanisms, which neutralize the abundant ROS generated under MGS-1 stress.

### 2.3. Nutritional Parameters

Similar to the results observed for stress markers, exposure to MGS-1 notably elevated the sugar levels in storage roots. Among the MGS-1 concentrations tested, the highest levels of sugar compounds were detected at 75% exposure. As compared to the control group, MGS-1 treatment resulted in fructose increases of 382.06%, 584.82%, and 695.86%; sucrose increases of 0.68%, 228.57%, and 650%; and glucose increases of 16.69%, 28.47% and 29.34% for 25%, 50%, and 75% MGS-1 exposures, respectively (Table 1 and Appendix A). The observed pattern of sugar increase—glucose > fructose > sucrose—in the storage roots suggests that plants treated with MGS-1 might exhibit heightened activity of the acid-soluble invertase enzyme, facilitating the conversion of sucrose into glucose and fructose, as compared to the control.

### 2.4. Amino Acid Profiling

Our study utilized the Partial least squares discriminant analysis (PLS-DA) to compare amino acid profiles. The score plot distinctly separated the control samples from those exposed to various MGS-1 levels, with PC1 accounting for 71.2% of the total variance (Figure 3A). This clear distinction underscores the profound impact of MGS-1 exposure on the amino acid profiles of storage roots. Notably, a 75% MGS-1 exposure greatly influenced the abundance of amino acids such as Gly, Ser, Phe, Lys, and Glu, each having a VIP score greater than one, highlighting their significance under MGS-1 stress. At 50% MGS-1 exposure, other amino acids, namely, Arg, His, Leu, and Pro, similarly exhibited VIP scores above one (Figure 3B). Detailed data on the differences in amino acid levels due to MGS-1 exposure can be found in Appendix A. Furthermore, to ascertain the relationship between the control group and various MGS-1 exposures, we validated the amino acid expression patterns using correlation coefficients and hierarchical cluster analysis. The control samples displayed a strong correlation with 25% (0.92) and 50% (0.83) MGS-1 exposures, a moderate correlation with the 75% exposure (0.5), and a weaker correlation (0.31) between the 25% and 75% exposure levels (Figure 3C). Hierarchical cluster analysis further confirmed these correlations, aligning with the findings from the correlation coefficient analysis (Figure 3D). Taken together, these analyses solidify the notion that MGS-1 exposure profoundly influences amino acid composition in storage roots.

### 2.5. Elemental and Total C:N Analysis

The inclusion of MGS-1 in the growth medium had marked effects on the mineral uptake and accumulation in both vines and storage roots as compared to control plants. Under MGS-1 exposure, there was a significant increase in the accumulation of various elements such as Al (up to 204.16%), B (up to 82.64%), Fe (up to 1193.86%), K (up to 66.22%), Mg (up to 339.18%), Mn (up to 946.6%), Si (up to 13.07%), and Zn (up to 104.47%). In contrast, there was a discernible reduction in the accumulation of Na (up to 52.88%) and P (up to 63.97%) in both the vines and storage roots as compared to the control. Among the tested MGS-1 concentrations, K and Si reached peak accumulation at 25% exposure. Conversely, higher levels of B, Mg, Mn, and Zn were observed at 50% exposure. When considering bioaccumulation efficiency, the vines demonstrated higher rates of elemental accumulation as compared to their corresponding storage root samples, irrespective of the MGS-1 treatment (Figure 4). Elements generally play roles in diverse plant metabolic and cellular functions. Hence, the interference of MGS-1 exposure on the uptake of these vital elements could disrupt the plants’ physiological processes, potentially hindering their growth.

In this study, contrary to the results from the elemental analysis, MGS-1 exposure considerably diminished the total C:N ratio in both the vines and storage roots of the sweet potato plant. The highest-observed C:N ratio (60.06%) was found in the storage roots under control conditions. Nevertheless, the C:N ratio consistently decreased as MGS-1 exposure concentrations increased, regardless of tissue type. Specifically, the C:N ratio fell by 40.82%, 49.25%, and 47.23% in the storage roots and by 16%, 17.03%, and 34.35% in the vines at 25%, 50%, and 75% MGS-1 exposure, respectively (Table 1).

## 3. Discussion

Plants have always garnered attention due to how they respond and maintain homeostasis [20]. Duri et al. [21] extensively reviewed the effects of lunar and Martian regolith simulants on the support of plant growth. Unlike the lunar regolith, no “real” sample of Martian regolith has been brought back to Earth. Martian regolith lacks organic matter and contains high levels of salts as well as a toxic level of perchlorate [21]. The reactive nitrogen on earth is primarily derived from the mineralization of organic matter [22]. Thus, supplementing Martian regolith simulant with organic matter or inorganic fertilizer is necessary to support the growth of crop plants. Our current study also indicates the challenges associated with crop cultivation on Mars. Our findings highlight the inhibitory effects of MGS-1 on sweet potato growth and overall health, and it identifies a combination of Martian regolith simulant and soil that could sustain the growth of sweet potato. However, we would like to point out here that our study was carried out under the assumption that crop growth on Mars will be conducted in a closed environment that will resemble an Earth-like environment, including light conditions, atmospheric conditions (e.g., temperature and day–night length), and gravity.

The intriguing capability of plants to adapt and respond under varied environmental and growth conditions has always captivated the scientific community, especially botanists and agronomists [23]. Central to this adaptive mechanism is the modulation of photosynthetic pigments, which play a pivotal role in determining a plant’s physiological health and its efficiency in converting light energy into chemical energy [24]. In our study, the assessment of chlorophyll yielded significant insights. Notably, there was a pronounced reduction in the total chlorophyll content in plants subjected to MGS-1 exposure. One probable explanation for this decrement could be the nutrient imbalances and the inherent challenges presented by the MGS-1 soil, which might have adversely affected the chloroplast’s function and diminished photosynthetic efficiency. Chlorophyll, being the chief pigment involved in light absorption and subsequent energy conversion, can be utilized as an early indicator of stress. As highlighted by Glanz-Idan and Wolf [25] and further corroborated by the extensive research of Zhou et al. [26], such declines in chlorophyll content are not just isolated events. Nutrient deficiencies, combined with hostile soil parameters, can impair chloroplast integrity, subsequently impacting the efficiency and organization of light-harvesting complexes [27,28]. Understanding these intricacies becomes even more critical when envisioning the challenges of cultivating crops in extraterrestrial settings like Mars [29].

The intricate dynamics of plant responses to stress encompass a spectrum of molecular, physiological, and biochemical adaptations. Delving deeper into this complex interplay, our study reveals compelling revelations regarding the stress markers of sweet potato under MGS-1 exposure. One of the most pronounced observations in this study was the increased accumulation of proline in the leaves, mirroring findings from many scientific investigations on stress adaptation [30,31]. Proline, often heralded as a quintessential osmoprotectant, showcased an accumulation pattern that was directly proportional to the increasing concentrations of MGS-1 and the plant’s age. Such accumulations, which have been recurrently documented in various organisms, are emblematic of the plant’s endeavor to stabilize its cellular structures and mitigate oxidative imbalances [32,33]. Proline is also well-known as an antioxidant, a metal chelator, and as a cellular signaling molecule [34].

An augmented ROS concentration, while being an innate response to stressors, can be deleterious, orchestrating a cascade of oxidative damages at the cellular level which, if unchecked, can culminate in cell senescence or even death [35]. In this experiment, MGS-1 exposure led to significant enhancement in the ROS levels of sweet potato (Figure 2A,B). A recent transcriptome analysis revealed a differential expression of the genes implicated in mediating responses to different stresses, including high salt concentrations, increased metal concentrations, and elevated ROS levels, when the model plant *Arabidopsis thaliana* was grown in diverse lunar regoliths that had been brought back to Earth by Apollo space missions [36]. It is important to mention here that both the lunar and Martian regolith possess high salt and metal contents, hence, plants grown directly on these regolith simulants may experience salt and metal stresses [21]. The intricate interactions of cellular defense mechanisms were further exemplified by the surge in ascorbic acid content within sweet potato storage roots as they grappled with escalating MGS-1 exposures. Ascorbic acid is well-known as an enzyme cofactor implicated in photosynthesis and also as an antioxidant [37].

When probing the effects of MGS-1 exposure on sweet potato, our investigation unraveled pronounced alterations in the sugar content within the storage root tissues. This metabolic recalibration, as evidenced by our results, suggests a robust surge in the activity of acid-soluble invertase enzymes, which facilitate the conversion of sucrose into its constituent monosaccharides: glucose, and fructose [38]. Such an elevation in sugar levels, while crucial for energy and carbon skeletons, may also be emblematic of their roles as osmolytes, fortifying cells against the duress of stress [39]. However, sugars don’t merely play passive roles; their ascendance could also influence cellular dynamics. They serve as pivotal signaling moieties, orchestrating cellular machinations from modulating gene transcription to fine-tuning enzymatic cascades [40].

Venturing beyond sugars, our study delved deep into the amino acid tapestry of storage roots, unearthing profound shifts in their profiles under MGS-1’s influence. These findings suggest that environmental stress, such as that simulated by MGS-1 exposure, can induce increased the levels of certain amino acids. Such enhancements have been previously noted to bolster rice plants’ defenses against salt stress by stabilizing their metabolisms and reducing ROS levels [41]. Amino acids are intricate players in plant physiology, especially during stress [42]. Our observations revealed that certain amino acids, likely acting as sentinel molecules, showcased heightened fluctuations that potentially encapsulate the plant’s strategic metabolic response to the rigors of MGS-1. Furthermore, amino acids like proline, glutamate, and glycine have been extolled for their roles as osmoprotectants, safeguarding plants by helping to maintain osmotic equilibrium amidst environmental adversities [43].

In our exploration, the inclusion of MGS-1 into the growth medium unveiled profound shifts in mineral dynamics, with both vines and storage roots undergoing marked changes in elemental uptake. Remarkably, when evaluating bioaccumulation efficiency, the vines consistently exhibited higher rates of elemental accumulation than their corresponding storage root samples, irrespective of the MGS-1 treatment. This disparity in accumulation patterns between vines and storage roots underscores the complex dynamics of mineral distribution within plant tissues. Minerals and elements, often dubbed the plants’ lifeblood, anchor many physiological and biochemical pathways [44]. Thus, the shifts induced by MGS-1 exposure could ripple through these pathways, reshaping the plant’s metabolic and growth trajectories. Conversely, a particularly intriguing observation was the MGS-1-mediated modulation of the C:N ratio. Traditionally, a perturbed C:N ratio is emblematic of plants grappling with environmental duress or traversing nutrient-scarce landscapes [45]. This dynamic equilibrium between carbon and nitrogen is more than just a mere ratio; it is a window into the plant’s metabolic mechanism, offering glimpses of processes ranging from carbon fixation to nitrogen assimilation [46]. Moreover, the observed fluctuations in the C:N ratio, a barometer of plant metabolic health, warrant a deeper inspection. Such fluctuations could be symptomatic of broader metabolic recalibrations. These might encompass shifts in foundational pathways, such as the Calvin cycle, which are central to carbon fixation and nitrogen assimilation processes and are pivotal for protein synthesis and growth [47,48].

## 4. Materials and Methods

### 4.1. Mars Global Simulant (MGS-1)

To assess the impact of Martian regolith on sweet potato growth, tolerance, and nutritional parameters, the Mars Global Simulant (MGS-1) was chosen for this study. The newly formulated MGS-1 soil was preferred because it had been modified to be free of hazardous components, which distinguishes it from other commercially available Mars simulants [19]. The simulant was procured from the Exolith Laboratory located in Oviedo, FL, USA. All of the analytical-grade chemicals used in this research were sourced from Sigma-Aldrich, St. Louis, MO, USA.

### 4.2. Experimental Design

For our greenhouse study at the Bioplex facility of West Virginia State University, we procured and acclimatized healthy, uniformly-sized sweet potato slips of the Beauregard variety. After about 2 weeks, when they had developed new roots, we conducted a pot experiment using varying combinations of MGS-1 and potting mix: pure potting mix was used as a control, and blends of 25%, 50%, and 75% MGS-1 exposure based on preliminary phytotoxicity screenings were used for the rest. Throughout the study, we maintained conditions at 28–36 °C during the day and 20–28 °C at night, with a 16-h light/8-h dark cycle and a consistent soil moisture level of 65–75%. We harvested plants at intervals of 40, 80, and 120 days, with the storage root’s total fresh weight (FW) measured electronically at 120 days post-harvest.

### 4.3. Estimation of Total Chlorophyll

We quantified the total chlorophyll content in sweet potato leaf tissues following the method described by Strain et al. [49], albeit with slight modifications. Specifically, we collected 50 mg of fresh leaf tissues from each treatment. These were then homogenized in a solution of 80% (*v*/*v*) ice-cold acetone fortified with 1 mM KOH, using three times the volume of the solution to the tissue weight. After homogenization, the mixture was centrifuged at 16,000× *g* for 2 min. Total chlorophyll content was subsequently determined using a spectrophotometer which measured at wavelengths of 663 nm for chlorophyll a and 646 nm for chlorophyll b. The results are presented in mg/g FW.

### 4.4. Determination of Proline Content

The proline content was quantified using the acid ninhydrin method, referencing the procedure outlined by Shabnam et al. [50], though with minor modifications. We homogenized approximately 0.5 g of leaf tissues in 300 µL of ice-cold water. This mixture was then centrifuged at 16,000× *g* for 5 min. The resulting supernatants were carefully transferred to a new tube. We added 1.25% acid ninhydrin to these supernatants and then incubated the mixture in a water bath at 100 °C for 30 min. After incubation, the reaction mixture was allowed to cool to room temperature. Using a spectrophotometer, the absorbance of the mixture was measured at 520 nm. By referencing a proline standard curve, we determined the proline concentration. The final results are expressed in terms of µmol/g FW.

### 4.5. 3,3′-Diaminobenzidine (DAB) Staining

The accumulation of ROS in sweet potato leaf samples, induced by MGS-1, was determined by measuring the H_2_O_2_ levels using the 3,3′-diaminobenzidine (DAB) method [51]. Fresh leaf tissues were initially submerged in a freshly prepared DAB staining solution, which consisted of 1 mg/mL of DAB (adjusted to pH 3.0) with the addition of 0.05% *v*/*v* Tween 20 and 2.5 mL of 200 mM Na_2_HPO_4_. These tissues were then subjected to a vacuum infiltration process at pressures ranging from 300 to 400 psi for 5 min. Following this, both the staining solution and the leaf tissues were incubated in dark conditions for 4 h, with a gentle shaking at 90 rpm. After incubation, the staining solution was removed, and the samples were destained using a bleaching solution: a mixture of ethanol, acetic acid, and glycerol in a 3:1:1 ratio. This destaining process continued for 15 min at 95 °C. The decolorized leaf samples were then examined under a light microscope (Zeiss Stemi 2000—C, Oberkochen, Germany). The presence of brown spots on these samples indicated the accumulation of H_2_O_2_.

### 4.6. Sample Preparation for GC-MS and LC-MS

After 120 days of treatment, the control and MGS-1-treated storage root samples were isolated. They were then dried at 80 °C for 24 h. Subsequently, these oven-dried storage root samples were processed for various instrumental analyses. The storage root samples in powdered form were extracted using distilled water, then centrifuged and divided into two equal portions. One portion was designated for sugar analysis, while the other was allocated for amino acid quantitation. For the sugar analysis, the extracted supernatants were dried. They were then derivatized: initially with 75 μL of methoxyamine hydrochloride (40 mg/mL in pyridine) for 60 min at 50 °C and subsequently with 75 μL of MSTFA + 1%TMCS at 70 °C for 120 min, with a further 2-h incubation at room temperature. Before derivatization, an internal standard (10 μL of hentriacontanoic acid at 1 mg/mL concentration) was introduced to each sample. The chromatograms were obtained using a GC-MS system (Agilent Inc, Santa Clara, CA, USA) outfitted with a ZB-5MS capillary column (60 m × 0.32 mm I.D. and 0.25 μm film thickness) from Phenomenex, CA, USA. During the process, a 1 μL aliquot was injected with a split ratio of 10:1. The Mass Hunter Quantitative Analysis software (version B.08.00) (Agilent Inc., Santa Clara, CA, USA) was employed to evaluate the target peaks, as described by Salih et al. [52]. Authentic standards produced calibration curves spanning the range of 0.5–50 μg/mL. To ensure consistent comparisons across samples, all data were adjusted based on the internal standard in each chromatogram and the weight of the sample.

### 4.7. Amino Acid Profiling

The amino acid profile of the storage roots, under various MGS-1 exposure levels, was analyzed using Liquid Chromatography-Mass Spectrometry (LC-MS). The analysis employed the Vanquish system (TSQ Altis LC-MS/MS system, Thermo Scientific), utilizing a Hypersil GOLD column (2.1 × 150 mm, 1.9 μ). The set flow rate was 600 μL/min. The mobile phases comprised 0.1% formic acid in water (A) and 0.1% in acetonitrile (B). The gradient was established as follows: from 0 to 0.5 min - 0% B, from 0.5 to 3.5 min - 60% B, from 3.5 to 5.5 min - 100% B, and from 5.5 to 7.5 min - 0% B. The injection volume was set at 1 μL, and the column chamber temperature was maintained at 500 °C. Data collection was carried out in a positive SRM mode with a voltage of 3500 V. Peak integration and quantification were conducted using the Thermo TraceFinder (version 4.1, Thermo Fisher, CA, USA) software, as detailed by Qiu et al. [53].

### 4.8. Total C:N

The dried samples from both the vines and storage roots were analyzed for their total carbon (C) and nitrogen (N) content. The estimation was conducted using the Dumas method, as outlined in EN 13654-2. A Skalar Primacs SNC 100 analyzer (Skalar, Breda, the Netherlands) was utilized for this purpose, as documented by Analytical, [54]. The C:N ratio of the samples was calculated from the obtained values of the total C and N.

### 4.9. Elemental Analysis

The elemental analysis was performed as described by Ali et al. [55], with slight modifications. Approximately, 500 mg of the powdered plant samples were measured and then subjected to acid digestion using a microwave digester (MARS 6, CEM Corporation, Matthews, NC, USA). The digested mixture was subsequently filtered and diluted using metal-free HPLC-grade water. The concentrations of the targeted elements in the samples were determined using inductively coupled plasma mass spectrometry (ICPMS, X—series II, Thermo, MA, USA). Calibration curves were generated with authentic multi elemental standards (Sigma-Aldrich, St. Louis, MO, USA).

### 4.10. Statistical Analysis

All experiments were carried out in triplicate (n = 3). Data were subjected to one-way ANOVA, and mean values were compared using Tukey’s test. Analyses were performed using the SPSS software (version 20, SPSS Inc., www.spss.com).

## 5. Conclusions

This study explored the reactions of sweet potato upon being exposed to MGS-1, a substance simulating Martian soil. Its findings revealed myriad physiological, biochemical, and molecular alterations in the plant. This investigation represents a significant advancement in space biology, shedding light on the intricate relationship between MGS-1 and critical facets of plant life, encompassing growth, stress resilience, and nutrient uptake. Notably, our experimental findings unveil a compelling trend: the 25% exposure level of MGS-1 emerges as the most favorable condition for studying plant performance, demonstrating a harmonious synergy between the simulated Martian regolith and the plant under examination, which results in optimal adaptation, a robust stress tolerance, and enhanced storage root yield. Conversely, as exposure levels rise, particularly to 75%, we observe substantial impediments to plant growth as characterized by the inhibition of key biochemical processes that are essential for vitality and productivity. Current research underscores the importance for further investigation into the molecular and genetic mechanisms related to plant adaptation and the nutrient requirements in Martian conditions; this investigation will set the stage for the development of resilient crop varieties that are uniquely suited to the challenges and opportunities of Mars. Ultimately, this study emphasizes the critical role of soil composition in ensuring the success of future colonization endeavors in extraterrestrial environments. As space colonization progresses, it becomes imperative to investigate deeper into the dynamics influencing crop plants’ adaptation and nutrition in Martian-like conditions.

## Figures and Tables

**Figure 1 plants-13-00055-f001:**
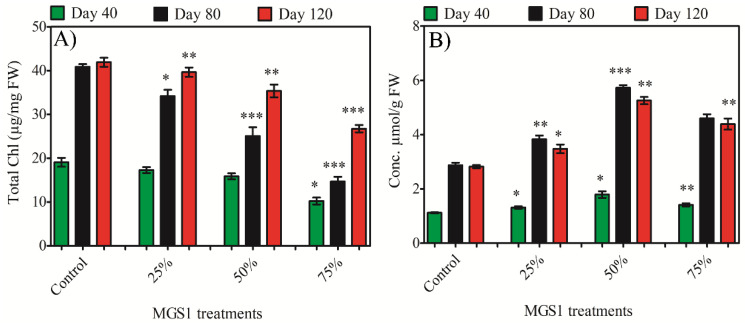
Effects of the different concentrations of MGS-1 exposure on (**A**) total chlorophyll and (**B**) proline of *I. batatas* leaves. Results are expressed as means of replicate ± SE, where ‘*’ indicates statistical significance (*p* < 0.05), ‘**’ indicates statistical significance (*p* < 0.01), and ‘***’ indicates statistical significance (*p* < 0.001) according to Tukey’s test.

**Figure 2 plants-13-00055-f002:**
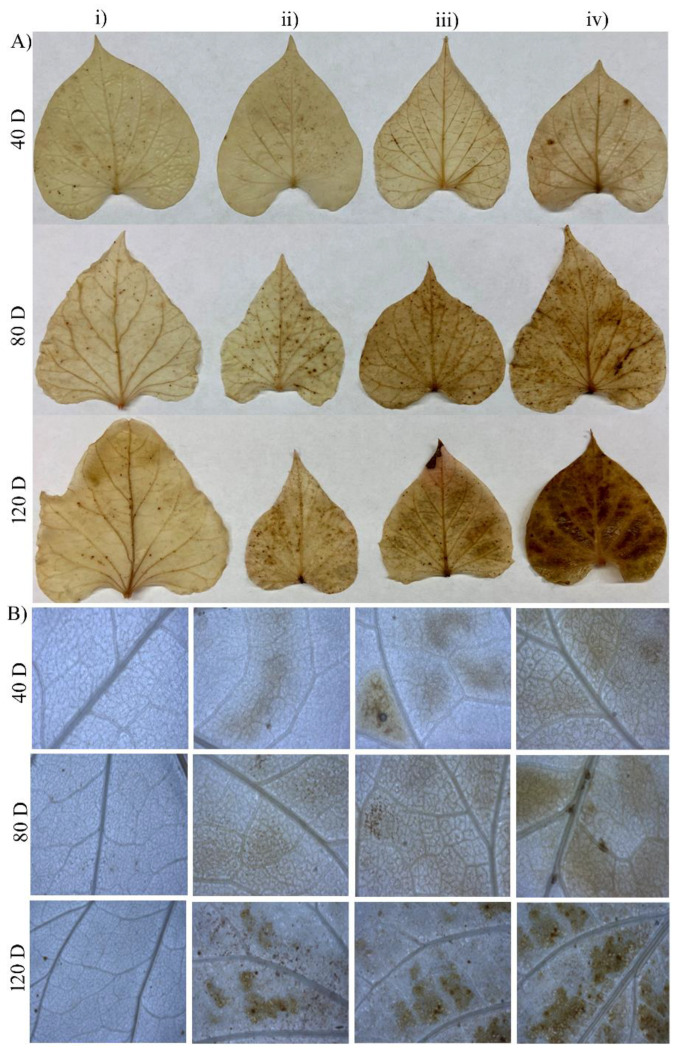
(**A**) Effects of various concentrations of MGS-1 exposure on H_2_O_2_ accumulation in *I. batatas* leaves, and (**B**) microscopic images of H_2_O_2_ accumulation. (i) control, (ii) 25%, (iii) 50%, and (iv) 75% of MGS-1 exposure.

**Figure 3 plants-13-00055-f003:**
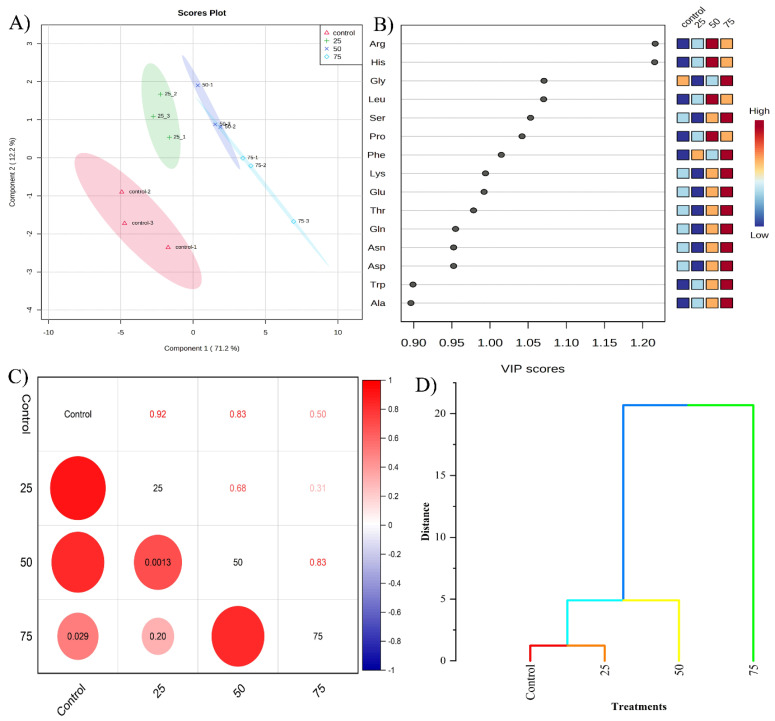
Amino acid profiling of storage root tissues grown under control and different levels of MGS-1 exposure. (**A**) PLS-DA score plot, (**B**) VIP plot, (**C**) correlation plot analysis, and (**D**) hierarchical cluster analysis.

**Figure 4 plants-13-00055-f004:**
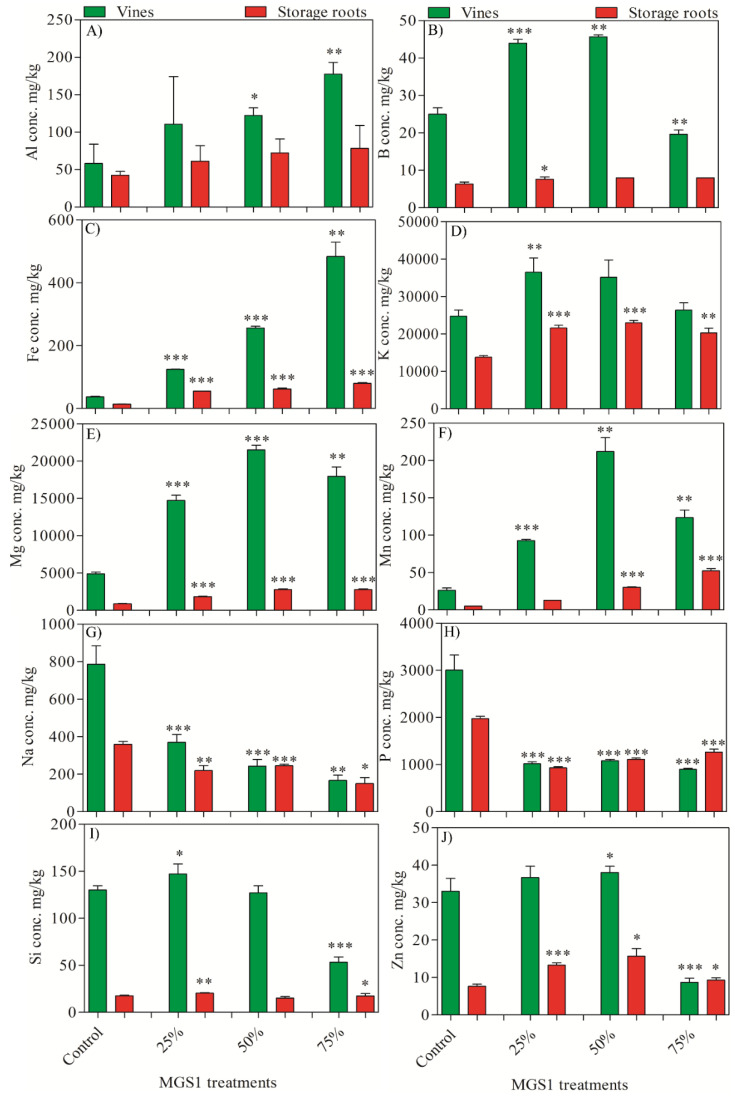
Effects of the different concentrations of MGS-1 exposure on the elemental accumulation in the vines and storage roots of *I. batatas* plants (**A**) Al, (**B**) B, (**C**) Fe, (**D**) K, (**E**) Mg, (**F**) Mn, (**G**) Na, (**H**) P, (**I**) Si, and (**J**) Zn. Results are expressed as means of replicate ± SE, where ‘*’ indicates statistical significance (*p* < 0.05), ‘**’ indicates statistical significance (*p* < 0.01), and ‘***’ indicates statistical significance (*p* < 0.001) according to Tukey’s test.

**Table 1 plants-13-00055-t001:** Effects of the different concentrations of MGS-1 exposure on the total sugar classes and the C:N ratio in *I. batatas*.

Treatments	Storage Root Biomass (gm/Plant)	Ascorbic Acid (µg/mg DW)	Sugar Parameters (µg/mg DW)			Total C:N Ratio (%)	
			Fructose	Sucrose	Glucose	Vines	SR
Control	99 ± 11.16	0.615 ± 0.04	0.145 ± 0.03	0.014 ± 0.001	10.36 ± 0.72	21.25 ± 0.34	60.06 ± 0.97
25% of MGS-1	59.4 ± 11.64 **	0.709 ± 0.06	0.699 ± 0.04 ***	0.015 ± 0.006	12.09 ± 1.02	17.85 ± 0.38 ***	35.54 ± 0.18 ***
50% of MGS-1	33.13 ± 6.66 **	0.838 ± 0.14	0.993 ± 0.03 ***	0.046 ± 0.002 ***	13.31 ± 0.46 **	17.63 ± 0.84 **	30.48 ± 0.04 ***
75% of MGS-1	6.66 ± 2.30 **	0.865 ± 0.08 *	1.154 ± 0.01 ***	0.105 ± 0.006 ***	13.4 ± 0.77 **	13.95 ± 0.13 ***	31.69 ± 0.48 ***

Note: DW—dry weight, SR—storage roots. Results are expressed as means of replicate ± SE, where ‘*’ indicates statistical significance (*p* < 0.05), ‘**’ indicates statistical significance (*p* < 0.01), and ‘***’ indicates statistical significance (*p* < 0.001) according to Tukey’s test.

## Data Availability

Data are contained within the article.

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
