# Peer review of "Effects of Mars Global Simulant (MGS-1) on Growth and Physiology of Sweet Potato: A Space Model Plant"

_plants, 2023, doi:10.3390/plants13010055_

Round 1
Reviewer 1 Report
Comments and Suggestions for Authors
This is an interesting study that also provides a detailed study of the changes in sweet potatoes under different MGS-1 ratios at the morphological, physiological, and molecular levels, which is of great significance for subsequent soil improvement and cultivation measures.
a con- 320 sistent soil moisture level of 65–75%,Please describe how to maintain the soil moisture?
Reviewer 2 Report
Comments and Suggestions for Authors
The manuscript presents a study of regeration, stress tolerance and dietary metrics of Ipomoea batatas, sweet potato, using Mars Global Simulant (MGS-1) concentrations (0, 25, 50, and 75%), analysing potato growth, storage root biomass, chlorophyll content, and also H2O2, proline, ascorbic acid, biochemical and aminoacids on plant tissues' . It is a quite interesting research, searching for new vegetal growth conditions with a very wide range of interventions in a very inhospitable environment full of heavy metals, sulfates and perchlorates.
Phytotoxicity effects were increasingly observed upon higher MGS exposure. Chlorophyll content was reduced significantly upon treatment with 50% or 75% MGS-1, for example.
The manuscript is very well written and the study also seems very well designed. The presentation model, however, in which the results are separated from their discussion, induces a mixture between the two items. For example, lines 157 to 160 present discussions of the newly entered results.
Paragraph 2.4 could be partitioned to bring figure 3 onto the same page and make the results easier to visualize.
At line 157, Figure 3B, not 4B.
Data from Figure 4 could be more discussed.
Observe the proper italics use for botanical names.
Comments on the Quality of English LanguageNone
Reviewer 3 Report
Comments and Suggestions for Authors
Dear Authors,
The manuscript received for review with the title „Effects of Mars Global Simulant (MGS-1) on Growth and Physiology of Sweet Potato: A Space Model Plant”, is of real interest regarding the unfriendly conditions on Martian soil for plant growth. The idea from which this research started is extremely important regarding the colonization of the planet Mars, even if it is now more theoretical than practical...unfortunately, the way in which this research was conducted, and the way of presentation is far from being published in a journal such as Plants-MDPI. I am not a fan of rejecting the manuscript, I appreciate the interest in solving such an important problem as ensuring a source of food in the conditions on Mars, unfortunately, it must be reviewed in its entirety.
Congratulations to the authors and I am waiting for the much-improved version of this manuscript.

Minor editing of English language required.
Round 2
Reviewer 3 Report
Comments and Suggestions for Authors
Unfortunately, the authors have only made small changes in terms of the English language...otherwise, no other scientific revision suggested in the first round of review.
Dear Authors,
The manuscript received for review with the title „Effects of Mars Global Simulant (MGS-1) on Growth and Physiology of Sweet Potato: A Space Model Plant”, is of real interest regarding the unfriendly conditions on Martian soil for plant growth. The idea from which this research started is extremely important regarding the colonization of the planet Mars, even if it is now more theoretical than practical...unfortunately, the way in which this research was conducted, and the way of presentation is far from being published in a journal such as Plants-MDPI. I am not a fan of rejecting the manuscript, I appreciate the interest in solving such an important problem as ensuring a source of food in the conditions on Mars, unfortunately, it must be reviewed in its entirety.
Congratulations to the authors and I am waiting for the much-improved version of this manuscript.

Author Response
Esteemed reviewer,
Please let us know where is the scope for the improvement rather than mentioning the manuscript has to undergo a major revision. Where exactly you are seeking the changes? Please be specific.
